# Modeling hepatitis C micro-elimination among people who inject drugs with direct-acting antivirals in metropolitan Chicago

Eric Tatara[1,2]☯*, Alexander Gutfraind[3,4], Nicholson T. Collier[1,2], Desarae Echevarria[3], Scott J. Cotler[3], Marian E. Major[5], Jonathan Ozik[1,2], Harel Dahari[3]*, Basmattee Boodram[6]☯*

**1** Consortium for Advanced Science and Engineering, University of Chicago, Chicago, Illinois, United States of America, **2** Decision and Infrastructure Sciences, Argonne National Laboratory, Argonne, Illinois, United States of America, **3** The Program for Experimental & Theoretical Modeling, Division of Hepatology, Department of Medicine, Stritch School of Medicine, Loyola University Chicago, Maywood, Illinois, United States of America, **4** Division of Epidemiology and Biostatistics, School of Public Health, University of Illinois at Chicago, Chicago, Illinois, United States of America, **5** Division of Viral Products, Center for Biologics Evaluation and Research, Food and Drug Administration, Silver Spring, Maryland, United States of America, **6** Division of Community Health Sciences, School of Public Health, University of Illinois at Chicago, Chicago, Illinois, United States of America

☯ These authors contributed equally to this work.
\* tatara@anl.gov (ET); hdahari@luc.edu (HD); bboodram@uic.edu (BB)

**Data Availability Statement:** The source code of the modeland all related tools can be accessed on the website [https://github.com/sashagutfraind/

## Abstract

Hepatitis C virus (HCV) infection is a leading cause of chronic liver disease and mortality worldwide. Direct-acting antiviral (DAA) therapy leads to high cure rates. However, persons who inject drugs (PWID) are at risk for reinfection after cure and may require multiple DAA treatments to reach the World Health Organization's (WHO) goal of HCV elimination by 2030. Using an agent-based model (ABM) that accounts for the complex interplay of demographic factors, risk behaviors, social networks, and geographic location for HCV transmission among PWID, we examined the combination(s) of DAA enrollment (2.5%, 5%, 7.5%, 10%), adherence (60%, 70%, 80%, 90%) and frequency of DAA treatment courses needed to achieve the WHO's goal of reducing incident chronic infections by 90% by 2030 among a large population of PWID from Chicago, IL and surrounding suburbs. We also estimated the economic DAA costs associated with each scenario. Our results indicate that a DAA treatment rate of >7.5% per year with 90% adherence results in 75% of enrolled PWID requiring only a single DAA course; however 19% would require 2 courses, 5%, 3 courses and <2%, 4 courses, with an overall DAA cost of $325 million to achieve the WHO goal in metropolitan Chicago. We estimate a 28% increase in the overall DAA cost under low adherence (70%) compared to high adherence (90%). Our modeling results have important public health implications for HCV elimination among U.S. PWID. Using a range of feasible treatment enrollment and adherence rates, we report robust findings supporting the need to address re-exposure and reinfection among PWID to reduce HCV incidence.

apk]. The software includes compiled binaries (available from https://zenodo.org/record/21714) that could be run on any system that supports Java 7 SE and simulated PWID databases to enable experimentation with APK. The authors are unable to provide direct access to the underlying PWID databases because they are considered Protected Health Information under institutional, state and federal laws. The Institutional Review Board of the University of Illinois (CNEP, YSN) and the Chicago Department of Public Health (NHBS) prohibit direct sharing of these data as they contain protected health information (PHS), including age and residence, about PWID.

**Funding:** This research is supported by the US National Institutes of Health (NIH) grants R01GM121600 and R01AI158666 and is based upon work supported by the US Department of Energy, Office of Science, under contract DE-AC02-06CH11357, and was completed with resources provided by the Research Computing Center at the University of Chicago (Midway2 cluster) and the Laboratory Computing Resource Center at Argonne National Laboratory (Bebop cluster). The funding sources had no role in the design of this study nor any role during its execution, analyses, interpretation of the data, or decision to submit results.

**Competing interests:** Nothing to disclose

# 1. Introduction

Hepatitis C virus (HCV) infection is a leading cause of chronic liver disease and mortality worldwide. Globally, an estimated 71 million people have chronic HCV infection, with an estimated 2.4 million in the United States [1], where the primary mode of HCV transmission is sharing syringes and other equipment among people who inject drugs (PWID). Fueled by the opioid epidemic, HCV incidence is rising, with 57,500 new cases in 2019 alone, a 63% increase from 2015 [2]. Access to and uptake of highly efficacious direct-acting antivirals (DAAs) for U.S. PWID remains low despite evidence supporting PWID can be successfully treated for HCV with sustained virologic response (SVR) similar to non-PWID [3]. Moreover, data from recent studies have shown that DAA therapy does not increase injection risk behaviors among PWID [4–6]; paradoxically, high uptake of DAA is expected to increase HCV incidence initially even with stable or decreased risk behaviors due to a temporary increase in the pool of PWID susceptible to reinfection [7]. DAA treatment is critical to achieving the World Health Organization's (WHO) goal of reducing incident chronic infections by 90% by 2030 [8]. As such, the effectiveness of treatment strategies on incidence should consider the impact of reinfection in PWID [9], particularly since drug use often spans decades with periods of temporary cessation [10].

Complex models that account for the effectiveness of DAAs on reducing new chronic HCV infections among PWID as well as the interplay of sociodemographic factors, risk behaviors and practices, social networks, and geographic location are needed to inform development of effective elimination strategies [11]. A recent review by Pitcher et al. [12], that includes more than 60 mathematical modeling papers, has provided some insight into HCV elimination strategies among PWID. In several studies, treatment was restricted to only once among PWID who failed to reach cure after DAA therapy [13, 14]. In particular, while Scott et al. [15] emphasized the importance of unrestricted treatment frequency, none of the previous modeling studies were designed to predict in detail the frequency of retreatment, the impact of retreatment on DAA cost, or the effect of treatment adherence on achieving the WHO goal.

A micro-elimination approach [16], which entails pursuing eliminations goals in discrete populations at high risk for transmitting HCV such as PWID, has been suggested as a less daunting approach that could build momentum by generating small victories towards achieving the WHO global HCV elimination goal. Even within the U.S., the PWID population is heterogeneous as evidenced by geographic differences in HCV incidence and prevalence [2, 17]. Using an agent-based model (ABM) approach, we focus on HCV micro-elimination among PWID in a targeted geographic region, metropolitan Chicago, Illinois (city of Chicago, Illinois and its surrounding suburban areas that encompass multiple counties) by examining the combination(s) of DAA enrollment (2.5%, 5%, 7.5%, 10%), adherence (60%, 70%, 80%, 90%) and number of treatments (1 to 4) needed to achieve the WHO's goal of reducing incident chronic infections by 90% by 2030 [18]. We also estimated DAA costs associated with each scenario.

# 2. Methods

## 2.1. HepCEP model synthetic population

We extended our previous work [18, 19] on simulating the PWID population in metropolitan Chicago, including the social interactions that result in HCV infection, to develop our Hepatitis C Elimination in PWID (HepCEP). The PWID population of metropolitan Chicago is heterogeneous and well-studied [20]. Details on the generation of the synthetic population was previously described [19] (S1 Table). In brief, parameter estimates were generated to profile each of the estimated 32,000 PWID [21] residing in metropolitan Chicago represented in the

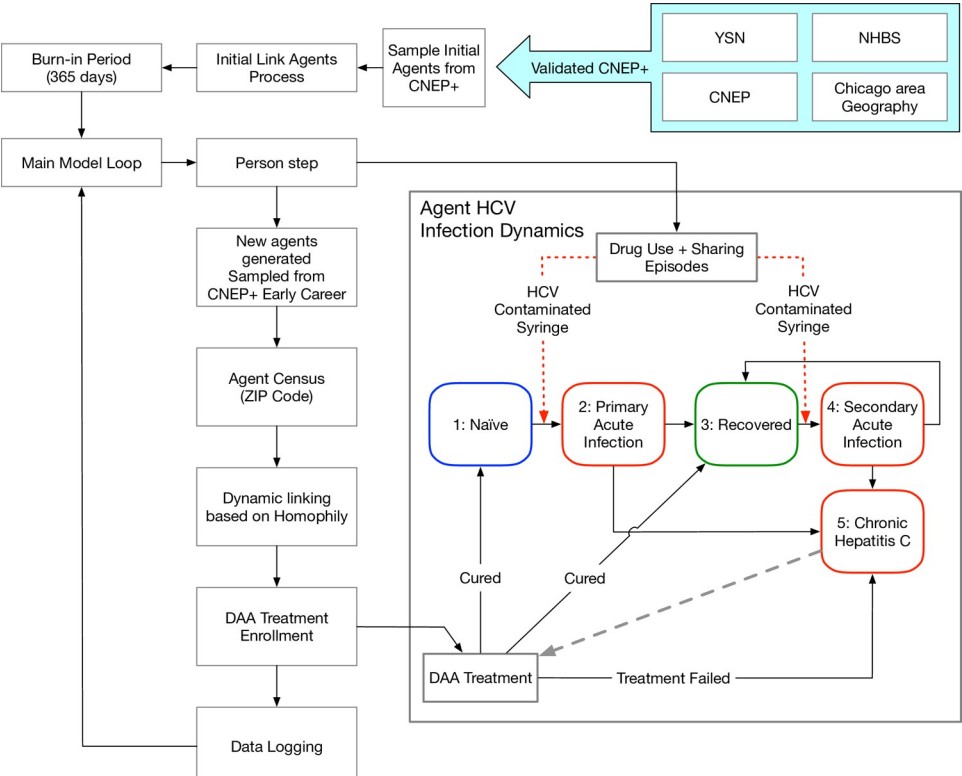

**Fig 1. Schematic diagram of the Hepatitis C Elimination in PWID (HepCEP) model.** The initial synthetic model population is generated from the CNEP+ dataset and linked in a syringe-sharing network. After the model burn-in period of 365 days, the main model loop begins and each individual PWID agent executes their step behavior that in turn simulates the HCV infection dynamics. PWID who have completed a successful treatment will return to either the NAÏVE or RECOVERED state depending on if they have previously recovered from an acute infection. When a PWID is cured, the model uses a CURED state but remembers past RECOVERED state or past NAÏVE state and returns the PWID to the respective state after treatment.

synthetic population [CNEP+] from analyses of two empirical datasets. These were the (i) 2009 metropolitan Chicago PWID data from the CDC-sponsored National HIV Behavioral Surveillance survey [22] of 545 PWID [NHBS 2009] and (ii) 2006–2013 data from a large, multi-site syringe service program (SSP) of >6,000 participants [20] [CNEP] (**Fig 1**). **Table 1** summarizes select attributes of the synthetic population, which mirrors some of the national pattern of PWID subgroups with the fastest increase in HCV incidence, including <30 years old and non-Hispanic white [2]. While national data and studies have shown large increases in HCV among non-urban populations, most of these focus on rural geographic areas. Our synthetic population includes under-studied suburban PWID, who comprise an estimated 54% of the metropolitan Chicago PWID population (**Table 1**). Suburban PWID present unique challenges to HCV elimination, including high levels of mobility between areas of high (Chicago) and low (suburb) HCV incidence areas and dispersed networks [23, 24].

## 2.2. Geographic environment and network formation

The metropolitan Chicago model geography is defined by zones based on the 2010 US Census ZIP code level data. Geographic locations of importance to PWID (residence, known drug market locations) from the two empirical datasets used to generate the synthetic population were embedded into the metropolitan Chicago geographic environment. Syringe-sharing was modeled as the primary mode of HCV transmission and PWID were connected via syringe-

**Table 1. Attributes of the synthetic population (CNEP+).**

| | |
|---|---|
| *Demographic attributes* | |
| Residence | Chicago: 46%; Suburbs: 54% |
| Race/ethnicity | Non-Hispanic (NH) white: 58%, Hispanic: 18%; NH-black: 21%; NH-other: 3% |
| Gender | Female: 30%; Male: 70% |
| Age | Mean: 35.3 years; IQR: 26.1–43.0; Over 30: 59%; Under 30: 41% |
| Enrollment in any SSP | SSP: 48%; non-SSP: 52%. |
| HCV infection state | Infected (acute or chronic): 30% |
| | Recovered (antibody +): 13% |
| *Behavioral attributes* | |
| Duration of injection drug use | Mean: 11.4 years; IQR: 3.3–16.0 |
| Probability of receptive sharing Ranges from 0 (never) to 1 (every injection) | Mean: 19%, IQR: 0%-37% |
| *Network attributes* | |
| In Degree (receptive network size) | 56% - 0 (no network), 32% - 1, 12%—≥2 |
| Out Degree (giving network size) | 65% - 0 (no network), 25% - 1, 10%—≥2 |

NH = Non-Hispanic; IQR = interquartile range; SSP = syringe service program

sharing networks (**Fig 2**). Network formation was determined by the probability of two persons encountering each other in their neighborhood of residence or within known drug market areas in Chicago, Illinois that attract both urban and non-urban PWID for drug purchasing and utilization of SSPs that are also located in the same areas [23]. The methods used to calculate network encounter rates, establishment processes, and removal of networks are detailed in [19]. Each individual has a predetermined number of in-network PWID partners who give syringes to the individual and out-network predetermined PWID partners who receive syringes from the individual, which drives the direction of HCV transmission. The network is dynamic, and during the course of simulation some ties may be lost, while new connections form, resulting in an approximately constant network size. PWID agents can leave the model population either due to age-dependent death or permanent drug use cessation and are replaced with new agents sampled from the input data set to maintain a nearly constant population size of 32,000 for the entire course of the simulation. The annual turnover rate of the population is about 2% (S1 Table).

## 2.3. Model validation

Two empirical datasets were obtained on metropolitan Chicago PWID to validate HepCEP. The 2012 NHBS Chicago PWID subset, the most representative data available at the time, was used to construct a synthetic population to validate HCV prevalence for 2012 (**Fig 1**). The previous validation results show high concordance, i.e., the predicted and actual values match within 2% overall for HCV prevalence. Similarly, data from a 2012–13 network and geographic study [25] of 164 PWID ages 18–30 and their drug-using network members, was used to calibrate and validate the network formation process. The simulated and actual networks match closely with an average error of 1.3% [19].

## 2.4. DAA treatment enrollment

Treatment enrollment was modelled as (unbiased) random sampling of chronically infected PWID and the annual target enrollment rate, defined as the total annual treatment enrollment

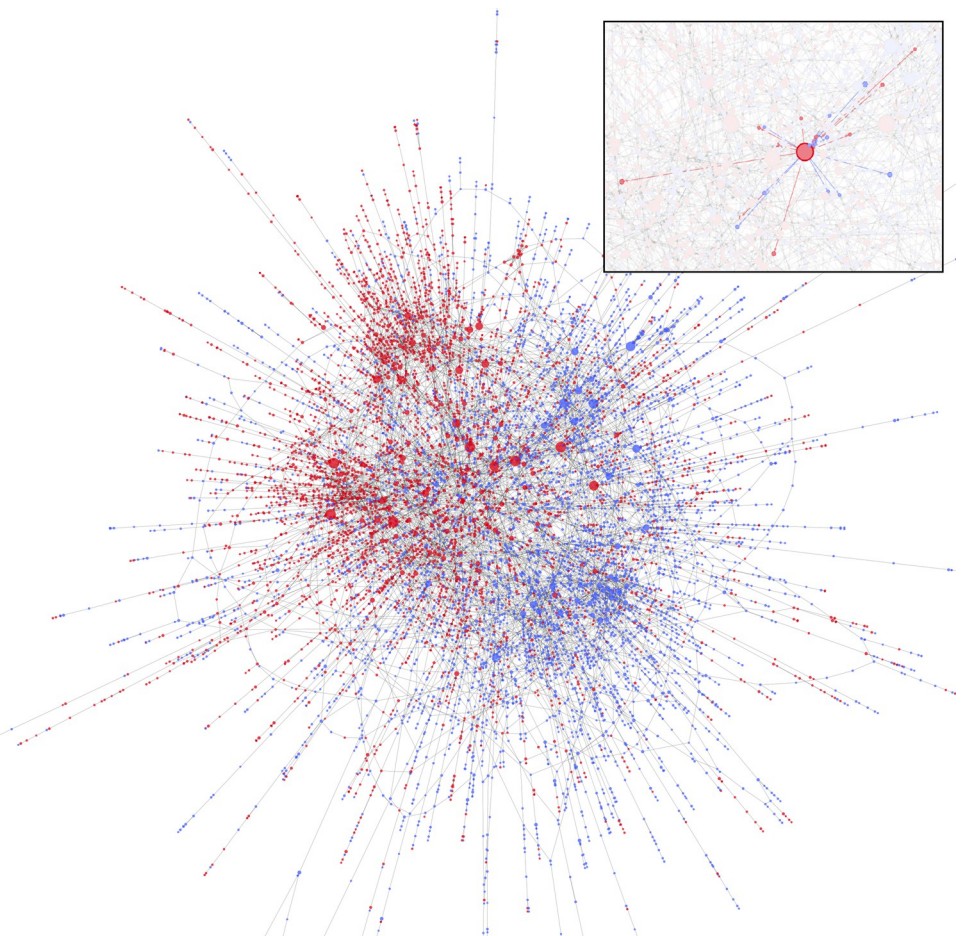

**Fig 2. PWID network visualization showing the syringe-sharing relationships between individual PWID in the synthetic population and colored by geographic location: Suburban (red) and urban (blue).** The number of individual PWID shown in this figure (9,731) represent 30% of the total PWID population who are part of the most highly connected section of the network and who have more than one network connection. The inset shows a single highlighted PWID and the individuals to who they are directly connected in the syringe-sharing network. Edge colors in the inset represent the locale of the recipient.

as a fraction of the total population, was a model parameter with a conservative range of 2.5–10%. DAA treatment success probability was a function of the treatment adherence and SVR parameters. While recently reported SVR rates are close to 99% [26–28] in many populations, we used a conservative estimate for SVR rates for U.S. PWID of 90%. The treatment adherence parameter was varied between 60%-90% to encompass the combined effects of behavioral, drug use and social factors that affect treatment completion (e.g., lost to follow-up, missed doses, enrollment in medication-assisted therapy, mobility) reported in the literature [23, 29–32]. Treatment re-enrollment(s) was allowed for PWID who successfully completed treatment and became re-infected. We assumed that successful treatments did not affect the probability of subsequent re-infections [7]. The total PWID target enrollment for a single day was determined by the daily mean treatment enrollment, which is the total PWID population multiplied by the annual treatment enrollment parameter / 365. The daily enrollment target was sampled from a Poisson distribution using the daily mean treatment enrollment. Other measures include treatment duration (12 weeks) and DAA cost ($25,000 [USD] per treatment) [33].

## 2.5. DAA treatment number

To examine the impact of DAA retreatments, we conducted a series of 80 different scenarios to account for all combinations of enrollment rate (2.5%, 5%, 7.5%, 10%), adherence (treatment completion) (60%, 70%, 80%, 90%), and a number of DAA treatment courses (e.g., 1–4; initial and three retreatments in response to up to four separate infections). For each of the 80 scenarios, 20 stochastic replicates were run in order to sufficiently capture variance in the model's output, for a total of 1,600 simulations. As such, we examined the impact of three retreatments after achieving cure or not achieving SVR after the treatment of the initial infection was performed to allow for examination of a retreatment policy that reflects reinfection frequency among PWID reported in published studies [34]. Although in clinical practice the number of DAA retreatments may be limited, to illustrate potential value of unconstrained retreatment policy on incidence, we also examined a scenario without re-treatment restrictions among those with reinfection and/or failed SVR.

## 2.6. ABM simulation timeframe

The ABM simulation start date of 2010 was selected based on the PWID demographic data from multiple surveys in previous years [19]. The model time step was one day, and treatment enrollment was started in year 2020 and ran until year 2030, with detailed model data collected on daily intervals. We report the mean annual incidence of chronic HCV relative to the mean baseline incidence rate in year 2020 with no treatment (enrollment rate of 0%). Each individual PWID agent steps through his current activity on each simulation day and transition between activities was dependent on the agent's current state (e.g. infected) and the scheduled duration of each activity.

The model Initialization and PWID agent behavior logic are shown in **Fig 1**. At the start of each simulation run, the initial synthetic model population is generated from the CNEP+ dataset and linked in a syringe-sharing network. The model is run with a "burn-in" period of 365 days used to stabilize the PWID network connectivity such that the number of syringe-sharing partners for each PWID converges to the predefined number of partners. After burn-in, the main model loop begins and each individual PWID agent executes his step behavior, which in turn simulates the HCV infection dynamics (**Fig 1**). Naïve PWID who are exposed to infected partners may develop a primary acute infection, which can either spontaneously clear or progress into a chronic infection (S1 Table). Recovered PWID who are again exposed to infected partners can be re-infected, and secondary acute infections can also clear or progress to a chronic infection (**Fig 1**).

PWID who have completed a successful treatment will return to either the Naïve or Recovered state depending on whether they have previously recovered from an acute infection (**Fig 1**). When a PWID is cured, the model uses a Cured state but remembers past Recovered state or past Naïve state and returns the PWID to the respective state after treatment. PWID who are Naïve and have never been in the Recovered state and who become infected enter the primary Acute stage and have a 65–88% chance (depending on gender) of entering the Chronic state, otherwise they enter the Recovered state [19]. PWID who are secondary Acute have previously been in the Recovered state (at any time) have an 85% chance of clearing and returning to the Recovered state, and 15% chance of becoming Chronic [19] (S2 Table).

**Fig 3** shows a schematic of the HepCEP model that highlights the activity timeline for a single PWID agent during model simulation and illustrates the detail and discrete nature of the model. **Fig 3** is an example timeline produced from a real simulation event log for which the total number of treatments is limited to four (i.e. initial and 3 retreatment courses). The frequency and timing of reinfection events is consistent with those reported in the literature for

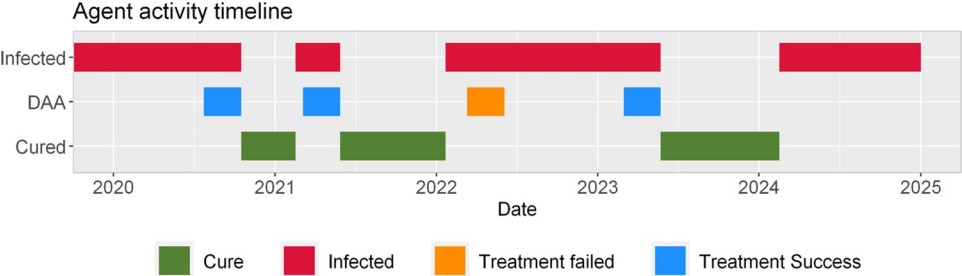

**Fig 3. Activity timeline for a single agent in the HepCEP model who was allowed only four courses of DAA therapy.** The colored bars indicate activities in which the agent is participating during the dates along the bottom of the timeline. The activity pattern shown in the figure are typical in some of HCV-positive agents that are selected for DAA treatment, cured, and re-infected multiple times. In this example, the agent was allowed to re-enroll in DAA treatment 3 times (total of 4 treatment courses), had a single occurrence of failed DAA treatment in year 2022 (orange bar) and eventually was re-infected ~1 year after SVR (in 2024) and remained chronically infected until 2030 (not shown).

PWID [34, 35]. The HCV-infected agent completes the following sequence (**Fig 3**): (i) enrolled in DAA treatment in mid-2020 and successfully cured, (ii) re-infected in 2021, retreated, and remained cured until early 2022, (iii) re-infected in early 2022, followed by a failed treatment in 2022, (iv) re-treated and cured again in late 2023, and (v) re-infected again in early 2024, but remained in the infected state until the end of the simulation in 2030 as the retreatment threshold of total 4 treatment courses (including one failed treatment course) has been exceeded. In the HepCEP model, individual PWID agent treatment can be customized on an individual level, allowing for treatment approaches and constraints to be uniquely set for each person.

## 2.7. DAA cost analysis of HCV elimination

With the cost of $25,000 per course of DAA treatment, we also estimated the overall DAA cost to achieve the WHO goal among PWID in metropolitan Chicago with select optional scenarios as determined by the study results (e.g., high adherence and up to three retreatments).

## 2.8. Simulation execution

Simulations were conducted using a high-performance computing workflow implemented with the EMEWS framework [36]. The simulation experiments were executed on the Bebop cluster run by the Laboratory Computing Resource Center at Argonne National Laboratory. Each simulation required approximately one hour of wall time to complete. Using the EMEWS workflow on the Bebop cluster, the actual compute time was also one hour since all runs can execute in parallel on 1,600 processes.

## 3. Results

**Fig 4** depicts chronic infection incidence for the four DAA-therapy enrollment rates when only a single DAA treatment course is permitted and assuming a treatment adherence of 90%. Due to the increasing availability of PWID cured with treatment who remain susceptible and can re-acquire HCV, there is a projected increase in incidence during the first 1–3 years after DAA therapy initiation, followed by a transient decline, then convergence to half of the incidence prior to DAA therapy initiation. This pattern does not achieve the WHO goal by year 2030, not even with DAA enrollment rates of up to 10% and treatment adherence of 90% (**Fig 4**). Since the WHO goal could not be achieved with a 90% treatment adherence with no

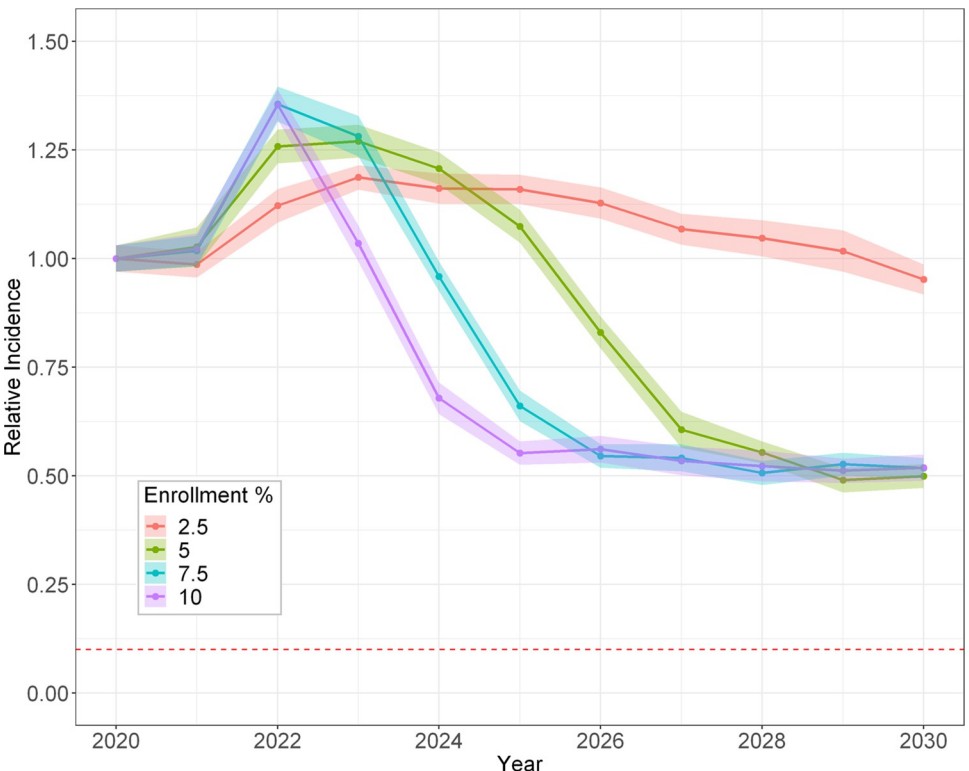

**Fig 4. Projected mean incidence of new HCV chronic infections among PWID relative to the predicted 2020 incidence with only one treatment course allowed.** Enrollment percent is DAA rate (e.g., enrollment of 10% is treatment of 100 per 1000 PWID per year) and treatment adherence of 90%. The ribbons represent the 95% confidence interval around the mean of 20 simulation runs. The horizontal red dashed line represents the WHO 2030 goal of 90% reduction in the incidence rate.

retreatment, no further simulations were conducted allowing lower treatment adherence rates with no retreatment.

**Fig 5** summarizes our simulation with unrestricted DAA treatment courses permitted (i.e, all infections and reinfections treated during the course of the simulation), with varying enrollment rates and treatment adherence levels. Overall, an enrollment rate of ≥5% with a treatment adherence threshold of ≥80% would be needed to achieve the WHO target of 90%

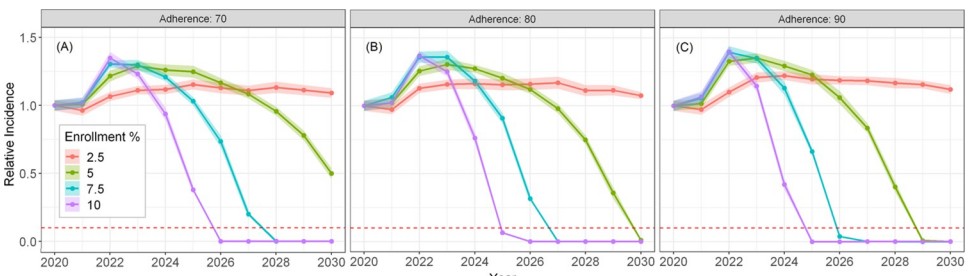

**Fig 5. Projected HCV mean incidence of new chronic infections among PWID relative to the predicted 2020 incidence during with no restriction on DAA treatment frequency.** Enrollment percent is DAA rate (e.g., enrollment of 10% is treatment of 100 per 1000 PWID per year) and treatment adherence of 70% (A), 80% (B), and 90% (C). The ribbons represent the 95% confidence interval around the mean of 20 simulation runs. The horizontal red dashed line represents the WHO 2030 goal of 90% reduction in the incidence rate.

**Table 2. Mean PWID treatment enrollment frequency and DAA costs (95% CI) for DAA treatment rate of 7.5% per year with unrestricted DAA courses permitted and a treatment adherence (TA) of 90% (A) and 70% (B).** Number treated values are rounded to the nearest integer. Percent treated is the fraction of PWID treated by number of times in each row relative to the total number of all individual PWID treated. DAA cost per treatment is $25,000.

| (A) TA 90% | | | | |
|---|---|---|---|---|
| Times Treated | Number of PWID Treated | | Percent | Cost [1K $] |
| 1 | 7368 | (7330–7406) | 75.4 | 184,201 (183,254–185,148) |
| 2 | 1805 | (1785–1826) | 18.5 | 90,273 (89,254–91,291) |
| 3 | 461 | (447–476) | 4.7 | 34,586 (33,505–35,668) |
| 4 | 108 | (104–113) | 1.1 | 10,825 (10,365–11,285) |
| 5 | 28 | (25–30) | 0.3 | 3,450 (3,088–3,812) |
| 6 | 5 | (4–6) | 0.1 | 803 (630–975) |
| 7 | 1 | (1–2) | < 0.1 | 256 (186–326) |
| 8 | 1 | (-) | < 0.1 | 229 (159–298) |
| 9 | 1 | (-) | < 0.1 | 225 (-) |
| **Total:** | **9777** | **(9739–9816)** | **100.0** | **324,395 (322,785–326,005)** |
| (B) TA 70% | | | | |
| Times Treated | Number of PWID Treated | | Percent | Cost [1K $] |
| 1 | 5773 | (5707–5761) | 58.5 | 143,324 (142,634–144,014) |
| 2 | 2382 | (2362–2403) | 24.3 | 119,123 (118,095–120,149) |
| 3 | 1005 | (988–1023) | 10.3 | 75,401 (74,084–76,718) |
| 4 | 413 | (402–424) | 4.2 | 41,330 (40,238–42,422) |
| 5 | 162 | (154–170) | 1.7 | 20,275 (19,254–21,296) |
| 6 | 64 | (60–68) | 0.6 | 9,555 (8,984–10,126) |
| 7 | 26 | (23–29) | 0.3 | 4,568 (4,091–5,044) |
| 8 | 10 | (8–11) | 0.1 | 1,920 (1,639–2,201) |
| 9 | 4 | (3–4) | < 0.1 | 844 (677–1,011) |
| 10 | 2 | (1–2) | < 0.1 | 417 (282–552) |
| 11 | 1 | (0–2) | < 0.1 | 367 (131–602) |
| 12 | 1 | (-) | < 0.1 | 300 (-) |
| 13 | 1 | (-) | < 0.1 | 325 (-) |
| **Total:** | **9,801** | **(9761–9841)** | **100.0** | **416,840 (414,371–419,309)** |

incidence reduction (Fig 5B and 5C); as such, an enrollment rate of 7.5% is a conservative lowest enrollment rate for which the WHO goal can be achieved by 2030 assuming an adherence of ≥70%. A DAA enrollment rate of 10% and a treatment adherence of 90% would achieve the WHO goal the earliest (year 2025, **Fig 5**C). As expected, adherence impacts the speed at which the WHO goal is met at lower enrollment rates. Overall, the HCV incidence rate reduction for this scenario demonstrates that the WHO goal is achievable by year 2030 when unrestricted DAA courses are allowed. However, while each PWID could potentially be treated without restriction in the model in this scenario, we show that only a small proportion of PWID require more than three DAA treatments for the duration of the simulation.

**Table 2** shows the frequency of how many times each PWID is treated using a treatment adherence of 90% (**Table 2**A) and 70% (**Table 2**B) and an enrollment rate of 7.5% when unrestricted DAA courses are allowed. As seen, 75.4% of PWID required only a single treatment and 18.5% of PWID require only two DAA treatments during the simulation period with 90% adherence (Table 2A), i.e., nearly 94% of all PWID require at most a total of two DAA treatments to meet the WHO goal by year 2030. Even with a treatment adherence of only 70%, nearly 93% of PWID require at most three DAA treatments (**Table 2**B). The frequency of retreatments in **Table 2** suggests that, for scenarios in which DAA treatment courses are

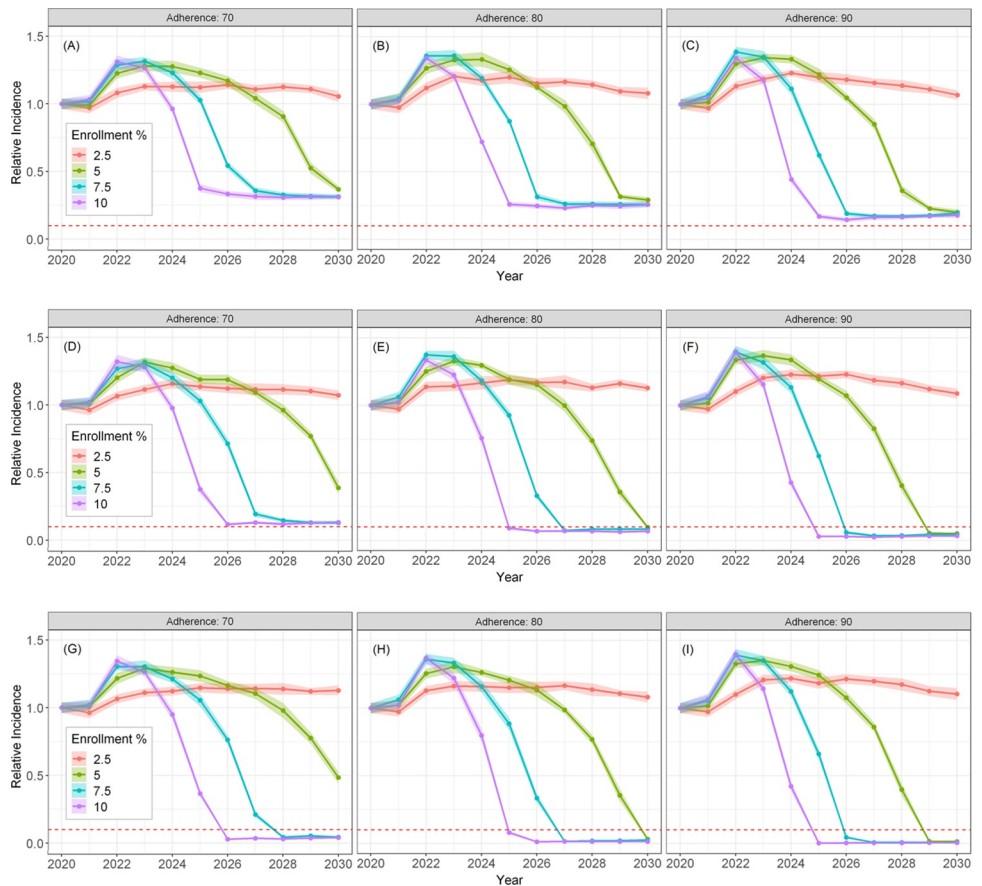

**Fig 6.** Projected HCV mean incidence of new chronic infections among PWID relative to the predicted 2020 incidence for 1–2 DAA treatment courses permitted (A-C), 2–3 DAA treatment courses permitted (D-F), and 1–4 DAA treatment courses permitted (G-I). Treatment adherence ranges from 70%-90% as indicated by the figure titles. The ribbons represent the 95% confidence interval around the mean of 20 simulation runs. The horizontal red dashed line represents the WHO 2030 goal of 90% reduction in the incidence rate.

unrestricted for each PWID, the actual fraction of the treated population requiring more than three treatments is only around 6% even when treatment adherence is low.

The incidence curves of new chronic HCV infections when DAA courses are limited to two (**Fig 6**A–6C) or three (**Fig 6**D–6F) are qualitatively similar to those for the unrestricted retreatment scenario in **Fig 5** such that the rate of incidence reduction was proportional to the DAA enrollment rate. However, the scenario in which up to two DAA courses are permitted (initial plus one retreatment) predicted that the WHO goal cannot be achieved even with the highest enrollment and adherence rates (**Fig 6**C). When increased to three (one initial plus two retreatments), the WHO incidence reduction goal is achievable by year 2030 for DAA enrollment rates $\geq$ 7.5% and adherence $\geq$ 80% (**Fig 6**E and 6F). As in the two DAA course scenario, the three DAA course scenario exhibits a lower limit on incidence reduction, although the limit approached close to zero for high treatment adherence rates ($\geq$ 90%) (**Fig 6**F). When up to four DAA courses are allowed per PWID (**Fig 6**G–6I), the incidence reduction goal was achieved for DAA enrollment rates $\geq$ 7.5% and adherence $\geq$ 70%, similar to the no treatment restriction scenario (**Fig 5**). This is explained by the very small fraction (<2%) of PWID requiring more than four DAA courses in the no treatment restriction scenario (**Table 2**),

**Table 3. Mean PWID treatment enrollment frequency (95% CI) for DAA treatment rate of 7.5% per year with 1–3 (Treatment Limit = 3) or 1–4 (Treatment Limit = 4) DAA courses permitted, and treatment adherence (TA) of 90% (A) and 70% (B).** Number treated values are rounded to the nearest integer. Percent treated is the fraction of PWID treated by number of times in each row relative to the total number of all individual PWID treated. The related DAA cost is shown in Table 4. Entries marked with (*) indicates scenario does not achieve WHO incidence elimination goal.

| (A) TA 90% | | | | | |
|---|---|---|---|---|---|
| | Treatment Limit = 3 | | | Treatment Limit = 4 | |
| Times Treated | Number of PWID Treated | | Percent | Number of PWID Treated | | Percent |
| 1 | 7291 | (7257–7325) | 74.3 | 7338 | (7303–7374) | 75.1 |
| 2 | 1808 | (1788–1828) | 18.4 | 1810 | (1787–1833) | 18.5 |
| 3 | 717 | (694–740) | 7.3 | 465 | (454–475) | 4.8 |
| 4 | -- | -- | | 165 | (157–173) | 1.7 |
| **Total:** | **9816** | **(9773–9860)** | **100.0** | **9,778** | **(9738–9818)** | **100.0** |
| (B) TA 70% | | | | | |
| Times Treated | Number of PWID Treated | | Percent | Number of PWID Treated | | Percent |
| 1 | 5577* | (5547–5607) | 56.2 | 5668 | (5641–5694) | 57.5 |
| 2 | 2340* | (2320–2360) | 23.6 | 2371 | (2354–2388) | 24.0 |
| 3 | 2003* | (1972–2033) | 20.2 | 1009 | (997–1022) | 10.2 |
| 4 | -- | -- | | 811 | (789–833) | 8.2 |
| **Total:** | **9,920***  | **(9875–9965)** | **100.0** | **9,859** | **(9821–9897)** | **100.0** |

suggesting that limiting the number of treatments per PWID to four was sufficient to achieve the WHO goal by 2030.

Table 3 provides a summary of the DAA treatment frequency for PWID in the up to three and up to four DAA course limit scenarios for low (70%, Table 3B) and high (90%, Table 3A) treatment adherence rates with a DAA enrollment rate of 7.5%. In the case of 90% adherence and a DAA enrollment rate of 7.5% with a treatment limit of three times, the model predicted that 74.3% of PWID required only a single treatment and 18.4% of PWID require only two DAA treatments during the simulation period; with a treatment limit of four times, 75.1% of PWID require only a single treatment, and 18.5% of PWID require at most two treatments (Table 3A).

## 3.1. DAA cost analysis for HCV elimination

The overall DAA cost to achieve the WHO goal among PWID in metropolitan Chicago with 90% adherence with up to four allowed DAA courses was predicted to be approximately $325.3 million (95% CI: 323.4–327.2, Table 4A), and nearly the same for restricting to up to three DAA courses ($326.4 million, 95%CI: 324.3–328.6, Table 4A). In comparison, at 70% adherence with four DAA courses restriction, the model predicted that substantially more DAA treatment courses (9859, 95% CI: 9821–9897, Table 3B) would be needed to achieve the WHO goal at a 28% increased cost ($417.0 million) (Table 4B). The difference is driven by treatment failure affecting the successful completion of the initial treatment (75% vs. 58%), thereby 2–3 times more PWID would need to re-enroll into treatment at the 90% compared to the 70% adherence scenario.

## 4. Discussion

The high cost of DAA treatment, challenges to adherence, and reinfection due to continued engagement in injection risk practices pose significant barriers to treatment access, uptake, and completion among PWID. In light of the heterogeneity of the U.S. PWID and variable regional [17] and subpopulation HCV incidence rates [2], our study aimed to elucidate

**Table 4. Mean treatment costs (95% CI) for DAA treatment rate of 7.5% per year with 1–3 (Treatment Limit = 3) or 1–4 (Treatment Limit = 4) DAA courses permitted, and treatment adherence (TA) of 90% (A) and 70% (B).** Cost values are rounded to the nearest 1K$. The DAA cost per treatment is $25,000. The related number of treatments is shown in **Table 3** Treatment costs for each group (times retreated) is calculated as the number treated in each group multiplied by the number of times treated multiplied by the cost per treatment. Entries marked with (*) indicates scenario does not achieve WHO incidence elimination goal.

| (A) TA 90% | | | | |
|---|---|---|---|---|
| | Treatment Limit = 3 | | Treatment Limit = 4 | |
| Times Treated | Cost [1K $] | | Cost [1K $] | |
| 1 | 182,278 | (181,435–183,120) | 183,458 | (182,569–184,346) |
| 2 | 90,395 | (89,399–91,391) | 90,485 | (89,336–91,634) |
| 3 | 53,790 | (52,084–55,496) | 34,845 | (34,028–35,662) |
| 4 | -- | -- | 16,520 | (15,710–17,30) |
| **Total:** | **326,463** | **(324,313–328,612)** | **325,308** | **(323,381–327,234)** |
| (B) TA 70% | | | | |
| Times Treated | Cost [1K $] | | Cost [1K $] | |
| 1 | 139,424* | (138,669–140,179) | 141,693 | (141,026–142,359) |
| 2 | 117,015* | (116,012–118,018) | 118,55 | (117,704–119,401) |
| 3 | 150,206* | (147,923–152,490) | 75,698 | (74,748–76,647) |
| 4 | -- | -- | 81,100 | (78,921–83,279) |
| **Total** | **406,645*** | **(404,050–409,240)** | **417,043** | **(414,619–419,446)** |

multiple pathways to HCV micro-elimination [16] among PWID from metropolitan Chicago with residents from both urban (46%) and suburban (54%) areas surrounding Chicago, Illinois (data not shown) [19]. Using realistic conservative enrollment rates (2.5% to 10%), we simultaneously examined the impact of adherence (70%-90%) and treatment frequency restrictions (unrestricted, <2, <3, <4 DAA courses per PWID) on reaching the WHO goal of reducing incident chronic HCV infection by 90%. Our results indicate that allowing treatment of reinfections is imperative regardless of enrollment and adherence and allowing for up to three DAA courses (**Fig 6**D–6F) is the minimum needed to achieve micro-elimination of chronic HCV infection incidence in this region. For PWID subpopulations with heightened challenges, e.g., low adherence due to residential transience and high levels of mobility, our study specifies a pathway to achieve the WHO target that includes a modest DAA enrollment rate of 7.5% (75 per 1000 PWID) per year, allowing for DAA treatment frequency of up to four times, and with a treatment adherence rate as low as 70% (**Fig 6**G–6I).

Total program costs for the scenario with 7.5% enrollment and 90% adherence were larger when multiple DAA courses were allowed compared to the scenario with a single course of treatment with no retreatment (**Table 5**). When retreatment was not considered as an active

**Table 5. Mean treatment costs, new chronic infections, and chronic reinfections (95% CI) during the treatment period (years 2020–2030) for DAA treatment rate of 7.5% per year and treatment adherence of 90%, by number of DAA courses allowed.** The DAA cost per treatment is $25,000. Cost values are rounded to the nearest 1K$ and infections are rounded to the nearest integer.

| Times Treated | Cost [1K $] | | Infections | | Reinfections | |
|---|---|---|---|---|---|---|
| 1 | 258,181 | (257,332–259,031) | 1725 | (1695–1756) | 92 | (87–97) |
| 2 | 318,556 | (316,675–320,437) | 1282 | (1248–1316) | 158 | (150–167) |
| 3 | 326,463 | (324,313–328,612) | 1096 | (1065–1127) | 112 | (106–119) |
| 4 | 325,308 | (323,381–327,234) | 1067 | (1042–1092) | 97 | (90–104) |

policy, the total DAA costs during the ten-year treatment enrollment period is $258.2 million (95% CI: 257.3–259.0), compared with a total cost of $325.3 million (95% CI: 323.4–327.2), for up to four allowed DAA courses (**Table 5**). However, as shown in our model, limiting DAA courses below three would not achieve the WHO goal for reducing new chronic infections by 2030 (**Fig 6**). Total costs between the scenarios that allow three ($326.4 million, 95%CI: 324.3–328.6, **Table 5**) compared to four ($325.3 million, 95% CI: 323.6–327.2, **Table 5**) courses were nearly identical. The counter-intuitive higher mean cost in the three versus four courses scenario can be partly attributed to stochastic variation in the model results, as the 95% CI for total cost overlap. However, perhaps more importantly, **Table 5** shows that the total number of infections during the DAA treatment enrollment period actually decreases as the number of allowed DAA courses increases. Limiting DAA treatment to a single course results in a larger pool of infected PWID that may infect other HCV-naïve PWID in their syringe-sharing network. Newly infected individuals would be subsequently enrolled in DAA treatment, incurring additional costs even though the PWID who is the source of the infection would be unable to re-enroll if treatment frequency limitations exist.

Our modeling results have important public health implications for HCV micro-elimination among U.S. PWID. Using a range of feasible treatment enrollment and adherence rates, we report robust findings supporting the need to address re-exposure and reinfection among PWID to reduce HCV incidence. Our ABM approach allows us to model PWID at the individual level and examine the effects of social network interactions on syringe-sharing and HCV transmission. In our recent ordinary differential equation (ODE) model study [37], we predicted that a DAA-treatment rate of 6.4%, with unlimited DAA treatments, and with an SVR rate of 90%, would be needed to reach the WHO elimination goal of 90% reduction of incidence over a 10-year treatment period with a total projected DAA cost of $418 million. This compares to a 7.5% DAA-treatment enrollment, 90% adherence rate using the HepCEP ABM with a lower DAA cost of $326.4 million if up to three DAA courses are allowed (**Fig 6**D–6F and **Table 5**), which would reach the WHO goal by 2026. The ODE approach does not represent the network structure or spatial and demographic heterogeneity of the PWID population that modulate the transmission risk and, therefore, results in an overestimate of the actual cost needed to reach >90% reduction. As such, our ABM is more suitable than ODE modeling for predicting the effects of any barriers to treatment.

Our study has several limitations. First, the model conservatively included both reinfection and unsuccessful treatment (failure to achieve SVR) in the single treatment only scenario (**Fig 4**), which effectively inflates the number of treatments needed to meet the WHO incidence goal. However, the WHO goal still cannot be achieved with an enrollment rate of 10% without treating reinfections. Second, our model assumes that PWID's underlying risk behaviors remain constant during the simulation, but patterns of drug use and injection risk practices may change over time (e.g., due to temporary cessation, medication-assisted therapy enrollment, etc.). Third, although 48% of the synthetic population were enrolled in SSP (**Table 1**), such that syringe-sharing and other HCV-related factors associated with SSP enrollment status are accounted for in the model, we did not directly evaluate the impact of scaling up harm reduction services in combination with other parameters. Fourth, DAA treatment was assigned in HepCEP randomly without considering the time and resources needed to screen and linkage to care, which may effectively extend the micro-elimination timeline. Fifth, PWID co-infected with HCV and HIV were not modeled in HepCEP; however, reported HIV prevalence was lowest among Chicago PWID (0.7%) of the 20 U.S. cities reporting to the 2018 NHBS (range 0.7–10.5%) [38]. Moreover, DAA treatment efficacy is reportedly comparable for HIV/HCV coinfected and HCV mono-infected patients [39–41]. As such, we do not expect HIV/HCV co-infected participants to have a major effect on the current ABM predictions.

A recent study suggests that the United States is not on track to meet the WHO goals for HCV elimination by 2030, with 35% of states, including Illinois, running behind by 10 years or more [42], which might be further delayed by the COVID-19 pandemic [43]. The study estimated an annual number of treatments of 1501–3000 (4.7%-9.4% of the estimated 32,000 PWID in metropolitan Chicago) would be required for Illinois to reach the treatment target for HCV elimination by 2030, which agrees with estimated optimal enrollment rates of >5%-10%. To address the lag in reaching the WHO elimination goals, it is imperative to implement strategies to increase HCV screening, linkage, and adherence to DAAs, and treatment of reinfections among PWID to achieve the WHO goal in high-risk populations.

## Supporting information

**S1 Table. Parameters for the generation of the synthetic population.**
(DOCX)

**S2 Table. HCV model infection parameters.**
(DOCX)

## Author Contributions

**Conceptualization:** Marian E. Major, Jonathan Ozik, Harel Dahari, Basmattee Boodram.

**Data curation:** Alexander Gutfraind, Basmattee Boodram.

**Formal analysis:** Eric Tatara.

**Funding acquisition:** Jonathan Ozik, Harel Dahari, Basmattee Boodram.

**Investigation:** Eric Tatara, Desarae Echevarria, Harel Dahari.

**Methodology:** Alexander Gutfraind, Nicholson T. Collier, Jonathan Ozik.

**Software:** Alexander Gutfraind, Nicholson T. Collier, Jonathan Ozik.

**Writing – original draft:** Eric Tatara, Marian E. Major, Jonathan Ozik, Harel Dahari, Basmattee Boodram.

**Writing – review & editing:** Eric Tatara, Alexander Gutfraind, Nicholson T. Collier, Desarae Echevarria, Scott J. Cotler, Marian E. Major, Jonathan Ozik, Harel Dahari, Basmattee Boodram.

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
