## [Decision Letter · Decision Letter 0]

22 Feb 2022

Modeling hepatitis C micro-elimination among people who inject drugs with direct-acting antivirals in metropolitan Chicago

PONE-D-22-00619

Dear Harel,

We’re pleased to inform you that your manuscript has been judged scientifically suitable for publication and will be formally accepted for publication once it meets all outstanding technical requirements.

Kind regards, and sincere congratulations !

Eve-Isabelle Pecheur

Academic Editor

PLOS ONE

Additional Editor Comments (optional):

Reviewers' comments:

Reviewer's Responses to Questions

**Comments to the Author**

1. Is the manuscript technically sound, and do the data support the conclusions?

Reviewer #1: Yes

2. Has the statistical analysis been performed appropriately and rigorously? 

Reviewer #1: Yes

3. Have the authors made all data underlying the findings in their manuscript fully available?

Reviewer #1: Yes

4. Is the manuscript presented in an intelligible fashion and written in standard English?

Reviewer #1: Yes

5. Review Comments to the Author

Reviewer #1: The background of the model is sufficient and clear in both introducing data bases and constructing the framework. The simulation method by using EMEWS is also effective to reflect comparable results. The cross-comparison between cases with varied enrollment rate, adherence and DAA treatment episodes provides sufficient results to detect the optimal strategy.

6. PLOS authors have the option to publish the peer review history of their article (what does this mean?). If published, this will include your full peer review and any attached files.

Reviewer #1: No

---

## [Editor Report · Acceptance letter]

28 Feb 2022

PONE-D-22-00619 

Modeling hepatitis C micro-elimination among people who inject drugs with direct-acting antivirals in metropolitan Chicago 

Dear Dr. Dahari:

I'm pleased to inform you that your manuscript has been deemed suitable for publication in PLOS ONE. Congratulations! Your manuscript is now with our production department. 

Kind regards, 

on behalf of

Dr. Eve-Isabelle Pecheur 

Academic Editor

PLOS ONE